# Factors Associated with Health-Related Quality of Life amongst Employees in a Malaysian Public University

**DOI:** 10.3390/ijerph182010903

**Published:** 2021-10-17

**Authors:** Mohd Rizal Abdul Manaf, Azmawati Mohammed Nawi, Noorlaili Mohd Tauhid, Hanita Othman, Mohd Rizam Abdul Rahman, Hanizah Mohd Yusoff, Nazarudin Safian, Pei Yuen Ng, Zahara Abdul Manaf, Nor Ba’yah Abdul Kadir, Kevina Yanasegaran, Siti Munirah Abdul Basir, Sowmya Ramakrishnappa, Mohd Izhar Ariff, Kurubaran Ganasegeran

**Affiliations:** 1Department of Community Health, Faculty of Medicine, Universiti Kebangsaan Malaysia, Kuala Lumpur 56000, Malaysia; azmawati@ppukm.ukm.edu.my (A.M.N.); rizam@ppukm.ukm.edu.my (M.R.A.R.); drhanie@ppukm.ukm.edu.my (H.M.Y.); nazarudin@ppukm.ukm.edu.my (N.S.); drsonygowda2007@gmail.com (S.R.); 2Department of Family Medicine, Faculty of Medicine, Universiti Kebangsaan Malaysia, Kuala Lumpur 56000, Malaysia; laili@ppukm.ukm.edu.my; 3Department of Pathology, Faculty of Medicine, Universiti Kebangsaan Malaysia, Kuala Lumpur 56000, Malaysia; drhanita@ppukm.ukm.edu.my; 4Drug and Herbal Research Centre, Faculty of Pharmacy, Universiti Kebangsaan Malaysia, Kuala Lumpur 50300, Malaysia; pyng@ukm.edu.my (P.Y.N.); ykevina@yahoo.com (K.Y.); 5Dietetic Program, Faculty of Health Sciences, Universiti Kebangsaan Malaysia, Kuala Lumpur 50300, Malaysia; zaharamanaf@ukm.edu.my (Z.A.M.); sitimunirah.abdulbasir@gmail.com (S.M.A.B.); 6Centre for Research in Psychology and Human Well-being, Faculty of Social Sciences and Humanities, Universiti Kebangsaan Malaysia, Bangi 43600, Malaysia; aknbayah@ukm.edu.my; 7Department of Medicine, Faculty of Medicine, Universiti Kebangsaan Malaysia, Kuala Lumpur 56000, Malaysia; izhar.ariff@ppukm.ukm.edu.my; 8Clinical Research Center, Seberang Jaya Hospital, Ministry of Health Malaysia, Penang 13700, Malaysia; medkuru@yahoo.com

**Keywords:** health-related quality of life, risk factors, employees, university, Malaysia

## Abstract

The current academic landscape has overwhelmed faculties and with demands to adopt tech-savvy teaching modes and accelerate scholarly works, administrative duties, and outreach programs. Such demands have deteriorated the health-related quality of life (HRQoL) among university employees. This study aimed to determine the factors associated with HRQoL among university employees in a Malaysian public university. This cross-sectional study was conducted among 397 employees from the Universiti Kebangsaan Malaysia (UKM) between April and June 2019. A self-administered questionnaire that consisted of socio-demographic items, risky health behaviors, health-related information, and validated scales for measuring employees’ physical inactivity, psychological states, and HRQoL was utilized. Descriptive and inferential statistics were calculated using SPSS version 23.0. Hierarchical multiple linear regression models were yielded to determine the factors associated with different domains of HRQoL. Mediation analysis was conducted using PROCESS MACRO (Model 4). Statistical significance was set to *p* < 0.05. Physical HRQoL scored the highest, while environmental HRQoL had the lowest score among the employees. Physical HRQoL was influenced by age, service duration, comorbid conditions, BMI, chronic diseases, and anxiety. Factors associated with psychological HRQoL were age, service duration, depression, and stress. Age, service duration, and chronic diseases affected employees’ social relationship HRQoL, while environmental HRQoL was associated with age, occupation type, chronic diseases, and depression. Socio-demographics, risky health behaviors, health profiles, and psychological attributes were significantly associated with employees’ HRQoL. Age was the only positively correlated factor across all HRQoL domains, while other factors deteriorated employees’ HRQoL.

## 1. Introduction

In the quest to foster specialized talents to meet current market employability, tertiary education plays a vital role in nurturing graduands of a high morale within societies, yet must consistently equip these future talents with relevant soft skills for marketability, up-to-date knowledge assimilation and their caliber of coping with job demands within the advancement of the scientific and technological landscape. Apart from the routine essential work required to be carried out by faculties, employees in these institutions are forced to excel as educators, scholars, mentors, or administrators [1]. Job demands such as continuous self-improvement in order to adopt revolutionary tech-savvy teaching, research, outreach programs, administrative or management work, and collaborations across institutions have indeed escalated faculty members’ dissatisfaction, causing their physical, psychological, and psycho-emotional wellbeing to be substantially compromised [2]. Although faculty members may sometimes perceive their vocation in their university working environment, tertiary education work life in current times can fundamentally be stressful, yet demanding, and deteriorates one’s health-related quality of life (HRQoL) [3].

HRQoL has been defined as “an individual’s perception of their position in life from the context of culture in which they live and in relation to their goals, expectations, standards and concerns [4].” It measures a person’s physical, psychological, and social relationships, as well as their relationships with salient features of their environment. Generic tools such as the WHOQOL-BREF [5] and SF-36 [6] are common instruments that are adopted to measure HRQoL across different populations, cultures, and settings. These tools tap dimensions that are relevant to exploring individuals’ personal characteristics; their wider use allows comparisons between groups across different study populations to establish consistencies and valid applications in order to determine outcome measures [7].

Studies that have evaluated the impact of work- and health-related factors on university employees’ HRQoL are scarce. The available literature postulated that reduced HRQoL among university employees had the tendency to greatly impact students’ lives and the people around them, thus compromising the quality of education and individuals’ daily routines [8]. Academic and non-academic staff in tertiary institutions were prone to high work pressure [9], causing them to be at risk for deteriorating physical health problems, such as headaches, obesity, hypertension, or cardiovascular diseases, as well as psychological effects, such as anxiety, stress, depression, or burnout [8,10,11,12].

It has been established that the HRQoL amongst faculty members is lower than in the general population [6]. Various factors are known to deteriorate university employees’ HRQoL. Chinese faculty members’ HRQoL was greatly influenced by socio-demographic (gender, age, education, marital status), occupational (work overload, job rank, working hours, research productivity), and health-related factors (having chronic diseases) [6]. One study evaluated the relationship between service duration and HRQoL among university employees [13]. While the literature has established a significant correlation between HRQoL and the body mass index (BMI) [14], studies that explored physical inactivity as a factor in the deterioration of university employees’ HRQoL showed mixed findings [2,15].

As universities fundamentally engage in the development of the society’s socio-cultural structure, the relationship between employees’ HRQoL and their performance is vital to institutions’ aims and successes [16]. Improvement of employees’ HRQoL has the capability of escalating the quality of teaching practices and deliveries [17], yet prompts a metric of success for an academic institution [18]. A better HRQoL among university employees catalyzes greater productivity, commitment, and performance quality in an academic institution [19]. As employees within tertiary institutions form the foundation of an organization’s structure and function within a university, there is a need to explore the different dimensions of HRQoL and their related factors to understand how these dimensions could risk influencing their performance in universities. With limited literature available in the context of an academic setting, the current study aimed to determine the factors associated with HRQoL among university employees from a Malaysian public university.

## 2. Materials and Methods

### 2.1. Study Design and Setting

An anonymous questionnaire-based survey was administered and a basic medical examination was conducted by the study team from the Universiti Kebangsaan Malaysia (UKM), Bangi, Selangor, Malaysia to address the study objectives. This was a descriptive–analytical cross-sectional single-institution study conducted between April and June 2019 via the simple random sampling method among employees of the UKM.

### 2.2. Sample-Size Determination

The required sample size for this study was determined based on a sample-size calculation for a finite population [20]. Using an approximate value of 4000 active employees at the university during the time of this study, a minimum sample size of 351 employees was calculated to represent a cross-section of the population and to allow the study to determine the HRQoL of employees with a margin of error of ±5%. An additional 20% was included in the calculated sample to compensate for missing data and non-response [21], for a final sample size of 421 employees.

### 2.3. Sample Selection

Four hundred and twenty-one employees from both academic and non-academic areas were randomly selected to participate in this study. Random selection of participants was conducted using the Research Randomizer Software [22]. The sampling frame included all active employees in the university at the time of this study, with their employment identity number provided by the Department of Registrar, Universiti Kebangsaan Malaysia. The single generated set of 421 random samples were identified, and subsequently, an invitation to participate in this study was sent to the official employee emails registered in the personal profiles in the university database. A date was given to each employee for data collection, which was conducted in the hall or foyer of selected faculties in the university. A set of self-administered questionnaires were administered to those who came on the day of data collection.

### 2.4. Inclusion and Exclusion Criteria

Permanent and contract employees from both academic and non-academic areas who were aged between 18 and 60 years were included in this study. Employees on maternity or sabbatical leave were excluded.

### 2.5. Study Instrument

A self-administered questionnaire that consisted of four parts was constructed for this study. The first part included the socio-demographic profile of the respondents (gender, age, marital status, occupation types, monthly household income level, and service duration). Occupation types were classified based on occupational grades as determined by the Public Education Service of Malaysia—academics (grades DS 41, 43, 44, and VK7), non-academic professionals (all categories 41 and above, except DS), and non-academic support staff (all categories below grade 41)—which were determined based on service duration, degree qualifications, and service performance for hierarchical promotional schemes [23,24]. Non-academic professionals were those with degree-level qualifications and above, but were not involved in academic teaching or learning activities. This occupational group mostly served as administrators or employees in the engineering, information, communication, and technology departments or units. In contrast, staff with qualifications below the degree level or those involved in support services, such as administrative duties (such as clerks or assistant officers), were classified as non-academic support staff [24]. Monthly household income was categorized according to the cut-offs of the Bottom 40% or B40 household group (lower-income households that earn less than MYR 4850 per month), Middle 40% or M40 household group (middle-income households that earn between MYR 4850 and 10,959 per month), and Top 20% or T20 household group (high-income households that earn equal to or more than MYR 10,960 per month) [25].

The second part of the questionnaire consisted of items on risky health behaviors and health-related information. The items included body mass index (BMI), smoking status, and being physically inactive. By using a portable stadiometer (SECA, Germany), employees’ height was measured while barefoot to the nearest 0.1 cm, while their weight was measured with a digital lithium weighing scale (Tanita, Japan) in light clothing and barefoot; it was calibrated to the nearest 0.1 kg. BMI was calculated based on a respondent’s body weight divided by the square of the height (kg/m^2^). BMI was later categorized based on the WHO BMI guideline (1998), which was also adopted in the National Health and Morbidity Survey (NHMS) of Malaysia of 2015 (<18.5 kg/m^2^ as underweight, 18.5–24.9 kg/m^2^ as normal weight, 25.0 to 29.9 kg/m^2^ as overweight, and ≥30 kg/m^2^ as obese) [26,27,28]. Current smokers were defined as respondents who had smoked at least 100 cigarettes during their lifetime and who currently smoked cigarettes [29]. The item was assessed using a dichotomized response (yes/no). To evaluate the physical inactivity of the respondents, the validated Malay version of the Global Physical Activity Questionnaire (GPAQ-M) was used [30]. The GPAQ-M is comprised of sixteen items that evaluate the intensity, frequency, and duration of participants’ physical activity across three dimensions—namely, physical activity at work, in travel or transport, and in recreation or leisure time, in addition to an extra item that collects data on sedentary behavior and time (minutes/day). Cut-offs were yielded: A metabolic equivalent task (MET) of 4 was categorized as moderate-intensity physical activity, and a value of 8 was categorized as vigorous intensity. The MET values were multiplied by the number of days per week of physical activity and the duration on a typical day for each dimension to compute the total physical activity levels of the respondents (MET-minutes/week). The high physical activity level was defined as vigorous-intensity activity on at least 3 days with at least 1500 MET-minutes/week or on seven days or more with any combinations of walking, moderate-intensity, or vigorous-intensity activities of at least 3000 MET-minutes/week. The moderate physical activity level was defined as 3 or more days of vigorous-intensity activity of at least 20 min/day, 5 or more days of moderate-intensity activity or walking for at least 30 min/day, or 5 or more days of any combination of walking, moderate-intensity, or vigorous-intensity activities to achieved a minimum of at least 600 MET-minutes/week. Respondents were classified as having a low physical activity level if they did not meet either of the two criteria above [31,32,33,34]. Health-related information, such as employees’ comorbid conditions (diabetes, hypertension, or hypercholesterolemia), was based upon self-reported measures as diagnosed by a medical doctor, or if they were currently under treatment with anti-hypertensives, oral hypoglycemics, or lipid-lowering drugs. Respondents were asked “Have you been diagnosed with diabetes, hypertension, or hypercholesterolemia by a medical doctor, or are currently under anti-hypertensive, oral hypoglycemic agents, or lipid-lowering drugs?”, with a response option of yes/no. Similarly, assessment of chronic diseases was based on a single item with a dichotomous response (yes/no) amongst respondents who had a medical history of stroke, myocardial infarction, cancer, or renal problems based on their self-reported measures as diagnosed by a medical doctor [35].

The third part of the questionnaire assessed perceived depression, anxiety, and stress among the respondents by using the validated Malay version of the 21-item Depression, Anxiety, and Stress Scale (DASS-21) [36]. The scale consisted of three subscales, each with seven items scored on a four-point Likert scale. The total scores for each subscale (depression, anxiety, and stress) were calculated by computing the scores for each item at their relevant subscales and multiplied by 2 to achieve a final score [37,38]. The results yielded for the perceived depression subdomain score were classified as normal (scores ranged between 0 and 9), mild depression (scores ranged between 10 and 13), moderate depression (scores ranged between 14 and 20), severe depression (scores ranged between 21 and 27), and extremely severe depression (scores of 28 and above). For perceived anxiety, scores were classified as normal (scores ranged between 0 and 7), mild anxiety (scores ranged between 8 and 9), moderate anxiety (scores ranged between 10 and 14), severe anxiety (scores ranged between 15 and 19), and extremely severe anxiety (scores of 20 and above). The total scores for perceived stress were classified as normal (scores ranged between 0 and 14), mild stress (scores ranged between 15 and 18), moderate stress (scores ranged between 19 and 25), severe stress (scores ranged between 26 and 33), and extremely severe stress (scores of 34 and above) [37]. To ease interpretation, the classified scores were subsequently dichotomized into normal and symptomatic (mild, moderate, severe, and extremely severe) for perceived depression, anxiety, and stress [39].

The main outcome measure, respondents’ HRQoL, was evaluated in the final part by using the validated Malaysian version of the WHOQOL-BREF questionnaire [40]. The 26-item five-point Likert scale questionnaire included one item from each of the 24 domains contained in the original WHOQOL-100, with two additional items on the overall HRQoL and general health. The 24 items were principally assembled into four major domains, namely, the physical HRQoL (7 items), the psychological HRQoL (6 items), social relationships HRQoL (3 items), and the environmental HRQoL (8 items). Three negatively worded items were reverse coded. The total computed scores for each domain were subsequently multiplied by 4 so that the scores were directly comparable with those derived from the WHOQOL-100 [5,35,40,41]. Higher domain scores indicated higher levels of HRQoL [40].

The face and content validity of the questionnaire was assessed by two experts; one was a psychologist who was also an author of this study (N.B.A.K.), while the other was an independent public health specialist who was not in the study team. The agreement on content between these two experts was concurrent. The questionnaire was subsequently pilot-tested among 20 employees who were not part of the study. Minor grammatical errors and misspellings were corrected, and the finalized version of the questionnaire was administered to participants. 

### 2.6. Statistical Analyses

Tests of the normal distribution of the total HRQoL subdomain scores and continuous variables were conducted. Descriptive statistics were reported for all variables. Student’s t-test and an ANOVA test (with post-hoc Bonferroni) were used to compare the means of HRQoL scores across demographics, risky health behaviors, health-related information, and psychological states of the respondents. Pearson correlation analysis was conducted to determine correlations between continuous variables. Hierarchical multiple linear regression analysis using the “Enter” technique was conducted to determine the factors that were significantly associated with HRQoL scores while simultaneously controlling for potential confounders and detecting probable mediators. All variables with *p*-values of less than 0.20 at the univariate level were selected for inclusion in the regression model. Demographic variables were entered in the first step, risky health behaviors were entered in the second step, health-related information was entered in the third step, and psychological states were entered in the fourth step. The model performance at each step was evaluated by yielding the model base and change statistics. Multicollinearity was checked between independent variables. To test for potential mediating effects in the association between predictors and HRQoL domains, PROCESS MACRO (Model 4) bootstrapped with 5000 samples was executed [42]. The bias-corrected–accelerated 95% confidence interval (BCa 95% CI) was estimated for each mediation (a × b product). A BCa 95% CI excluding 0 indicated a significant mediating role. Statistical significance was set to *p* < 0.05. The analysis was conducted using IBM SPSS Statistics Software version 23.0 [43].

## 3. Results

### 3.1. Sample Characteristics

Table 1 shows the sample characteristics. A total of four hundred and twenty-one employees were approached and 397 (response rate: 94.3%) participated. Twenty-four responses were excluded from the analysis due to missing data (15 responses) and withdrawal from the study (nine participants). Of the total employees, 250 (63%) were women and 147 (37.0%) were men. The mean (SD) age of the employees was 43 (7.6) years, and the age ranged between 27 and 60 years old. Most employees were married (353; 88.9%), were in the M40 household income group (141; 48.6%), and had more than 10 years of service (344; 86.6%). The majority of the employees were non-academic support staff (227; 57.2%). Most employees were overweight (159; 40.1%) and physically inactive (229; 58.6%). However, only thirty (7.6%) were smokers. Twelve (3.0%) of the employees suffered from chronic diseases, while thirty (7.6%) had comorbid conditions. Depressed, anxious, and distressed employees constituted 33.5%, 51.1%, and 24.4% of the sample, respectively.

The mean (SD) of the physical HRQoL score was 72.6 (12.9) and the score ranged between 31 and 100. The mean (SD) of the psychological HRQoL score was 71.5 (12.1) and the score ranged between 25 and 100. The mean (SD) of the social relationship HRQoL score was 72.4 (14.8) and the score ranged between 19 and 100. The mean (SD) of the environmental HRQoL score was 70.2 (11.9) and the score ranged between 38 and 100. The mean (SD) of the total HRQoL score was 99.4 (11.1) and the score ranged between 64 and 127 (Table 1).

### 3.2. Association between HRQoL and Sample Characteristics

Table 2 shows the association between the HRQoL domain scores and the socio-demographic characteristics of the employees. There was a statistically significant correlation between employees’ ages and psychological HRQoL scores (r = 0.119, *p* = 0.018). Statistically significant associations were observed between the household income level and psychological HRQoL score (*p* = 0.029); post hoc tests revealed that those within the M40 income group (70.4 ± 12.3) had lower psychological HRQoL scores in comparison to those within the T20 income group (74.9 ± 10.1, *p* = 0.024). Employees who were in service for more than 10 years had lower social relationship HRQoL scores (71.7 ± 14.8) in comparison to those in service for 10 years or less (77.0 ± 14.0, *p* = 0.015).

There was a statistically significant correlation between employees’ ages and environmental HRQoL scores (r = 0.206, *p* < 0.001). Statistically significant associations were observed between occupation type and environmental HRQoL score (*p* < 0.001); post hoc tests revealed that non-academic support staff had lower environmental HRQoL scores (68.0 ± 12.5) compared to non-academic professionals (72.9 ± 10.7, *p* = 0.005) and academics (73.4 ± 10.7, *p* = 0.001). Similarly, statistically significant associations were observed between the household income level and environmental HRQoL score (*p* = 0.032); post hoc tests revealed that those within the B40 income group had lower environmental HRQoL scores (68.7 ± 12.2) in comparison to those in the M40 income group (69.6 ± 11.4, *p* = 0.044) (Table 2).

### 3.3. Association between HRQoL and Risky Health Behaviors, Health-Related Information, and Psychological States

Table 3 exhibits the association between the HRQoL domain scores and risky health behaviors, health-related information, and psychological states. Statistically significant associations were observed between BMI and the physical HRQoL score (*p* = 0.002); post hoc tests revealed that those who were obese (69.0 ± 12.9) had lower physical HRQoL scores in comparison to those who were overweight (74.5 ± 12.4, *p* = 0.005). Employees with more than one comorbid condition (66.2 ± 14) had lower physical HRQoL scores compared to those with one or no comorbid conditions (73.0 ± 12.8, *p* = 0.006). Those with chronic diseases (64.3 ± 12.2) had lower physical HRQoL scores compared to those without chronic diseases (72.8 ± 12.9, *p* = 0.026). The physical HRQoL scores were lower amongst employees who were depressed (68.3 ± 13.7, *p* < 0.001), anxious (69.1 ± 12.5, *p* < 0.001), and distressed (68.8 ± 13.7, *p* = 0.001) in comparison to those with normal psychological states.

Statistically significant associations were observed between BMI and the psychological HRQoL score (*p* < 0.001); post hoc tests revealed that those who were obese had lower psychological HRQoL scores (67.8 ± 11.9) compared to those who were underweight (84.5 ± 6.8, *p* = 0.005), normal weight (72.3 ± 12.2, *p* = 0.024), and overweight (72.8 ± 11.5, *p* = 0.005). Psychological HRQoL scores were lower among employees who were depressed (66.6 ± 14.0, *p* < 0.001), anxious (69.4 ± 12.4, *p* < 0.001), and distressed (68.3 ± 13.9, *p* = 0.002) in comparison to those with normal psychological states (Table 3).

Those with chronic diseases (62.5 ± 18.1) had lower social relationship HRQoL scores compared to those without chronic diseases (72.6 ± 14.6, *p* = 0.019). Social relationship HRQoL scores were lower amongst employees who were depressed (68 ± 16.9, *p* < 0.001), anxious (69.9 ± 15.2, *p* < 0.001), and distressed (68.8 ± 17.1, *p* = 0.006) in comparison to those with normal psychological states (Table 3).

Statistically significant associations were observed between BMI and the environmental HRQoL score (*p* = 0.007); post hoc tests revealed that those who were obese (67.9 ± 11.7) had lower environmental HRQoL scores in comparison to those who were overweight (72.3 ± 12.3, *p* = 0.025). Environmental HRQoL scores were lower amongst employees who were depressed (67.1 ± 13.6, *p* < 0.001), anxious (67.7 ± 12.5, *p* < 0.001), and distressed (67.9 ± 13.8, *p* = 0.029) in comparison to those with normal psychological states (Table 3).

### 3.4. Factors Associated with Physical HRQoL by Hierarchical Multiple Linear Regression Analysis

Table 4 shows the factors associated with the physical HRQoL domain score according to the hierarchical multiple linear regression analysis. Twelve variables accounted for 14.2% of the total variance for the physical HRQoL domain score. The control variables in step 1 accounted for 3.1% of the variance in the physical HRQoL domain score (F change = 3.08), and among the variables, age (β = 0.186, *p* < 0.001) and service duration of more than 10 years (β = −4.761, *p* = 0.020) were significantly associated with the physical HRQoL domain score. The total variance explained increased slightly to 3.7% after including BMI in the second step, and the statistically significant variables associated with physical HRQoL remained unchanged, as in step 1. Adding comorbid conditions and chronic disease variables in the third step contributed to an additional 3.3% of the variance in the physical HRQoL domain score (F change = 6.828), with age (β = 0.257, *p* = 0.005), service duration of more than 10 years (β = −4.441, *p* = 0.027), and having more than one comorbid condition (β = −6.893, *p* = 0.006) being significantly associated with the physical HRQoL domain score. Finally, adding the psychological variables (depression, anxiety, and stress scores) contributed to an additional 7.2% of the variance in the physical HRQoL domain score. The final step in the model yielded age (β = 0.219, *p* = 0.015), being obese (β = −10.197, *p* = 0.045), having more than one comorbid condition (β = −6.621, *p* = 0.006), having a chronic disease (β = −7.237, *p* = 0.049), and being anxious (β = −6.128, *p* < 0.001) as significant factors associated with the physical HRQoL domain score.

### 3.5. Factors Associated with Psychological HRQoL by Hierarchical Multiple Linear Regression Analysis

Fourteen variables accounted for 25.5% of the total variance for the psychological HRQoL domain score. The control variables in step 1 accounted for 7.3% of the variance in the psychological HRQoL domain score (F change = 3.700), and among the variables, age (β = 0.338, *p* = 0.001) and service duration of more than 10 years (β = −4.972, *p* = 0.011) were significantly associated with the psychological HRQoL domain score. The total variance explained increased by 2.8% after including the BMI and physical inactivity variables in the second step, and the statistically significant variables associated with psychological HRQoL included age (β = 0.326, *p* = 0.001), service duration of more than 10 years (β = −4.503, *p* = 0.021), and being obese (β = −11.527, *p* = 0.048). Adding the chronic disease variable in the third step contributed slightly to an additional 0.3% of the variance in the psychological HRQoL domain score (F change = 0.885). The statistically significant variables associated with the psychological HRQoL domain score in this step were age (β = 0.329, *p* = 0.001) and service duration of more than 10 years (β = −4.386, *p* = 0.025). Finally, adding the psychological variables (depression, anxiety, and stress scores) contributed to an additional 15.1% of the variance in the psychological HRQoL domain score. The final step in the model yielded age (β = 0.269, *p* = 0.004), service duration of more than 10 years (β = −4.907, *p* = 0.007), being depressed (β = −9.842, *p* < 0.001), and being distressed (β = −6.600, *p* = 0.048) as significant factors associated with the psychological HRQoL domain score (Table 5).

### 3.6. Factors Associated with Social Relationship HRQoL by Hierarchical Multiple Linear Regression Analysis

Table 6 shows the factors associated with the social relationship HRQoL domain score according to the hierarchical multiple linear regression analysis. Twelve variables accounted for 10% of the total variance for the social relationship HRQoL domain score. The control variables in step 1 accounted for 3.2% of the variance in the social relationship HRQoL domain score (F change = 3.215), and among the variables, age (β = 0.231, *p* < 0.025) and service duration of more than 10 years (β = −6.364, *p* = 0.006) were significantly associated with the social relationship HRQoL domain score. The total variance explained increased to 5.2% after including the BMI and physical inactivity variables in the second step, and the statistically significant variables associated with the social relationship HRQoL remained unchanged, as in step 1. Adding the chronic disease factor in the third step contributed to an additional 1.4% of the variance in the social relationship HRQoL domain score (F change = 5.875), with age (β = 0.255, *p* = 0.013), service duration of more than 10 years (β = −5.769, *p* = 0.012), and having chronic diseases (β = −10.415, *p* = 0.016) being significantly associated with the social relationship HRQoL domain score. Finally, adding the psychological variables (depression, anxiety, and stress scores) in the fourth step contributed to an additional 3.4% of the variance in the social relationship HRQoL domain score. The final step in the model yielded service duration of more than 10 years (β = −4.911, *p* = 0.032) and having chronic diseases (β = −10.265, *p* = 0.016) as significant factors associated with the social relationship HRQoL domain score.

### 3.7. Factors Associated with Environmental HRQoL by Hierarchical Multiple Linear Regression Analysis

Twelve variables accounted for 20.0% of the total variance for the environmental HRQoL domain score. The control variables in step 1 accounted for 9.2% of the variance in the environmental HRQoL domain score (F change = 5.700), and among the variables, age (β = 0.334, *p* < 0.001) and being non-academic support staff (β = −4.479, *p* = 0.036) were significant factors associated with environmental HRQoL domain score. The total variance explained increased by 0.5% after including BMI in the second step, and the statistically significant variables associated with environmental HRQoL remained unchanged, as in the first step. Adding the chronic disease variable in the third step contributed slightly to an additional 1.3% of the variance in the environmental HRQoL domain score (F change = 3.962). The statistically significant variables associated with the environmental HRQoL domain score in this step were age (β = 0.345, *p* < 0.001), non-academic support staff (β = −4.513, *p* = 0.034), and having chronic diseases (β = −8.000, *p* = 0.048). Adding the psychological variables (depression, anxiety, and stress scores) in step 4 contributed an additional 9.1% of the variance in the environmental HRQoL domain score (F change = 10.403). The final step in the model yielded age (β = 0.317, *p* < 0.001), non-academic professionals (β = −4.128, *p* = 0.042), and being anxious (β = −4.948, *p* < 0.001) as significant factors associated with the environmental HRQoL domain score (Table 7).

### 3.8. Results of Mediating Effects

The association between service duration and physical HRQoL was attenuated and became non-significant when the psychological variables (depression, anxiety, and stress) were added into step 4 of the regression model, suggesting the presence of mediating effects. Similarly, the associations of age with social relationship HRQoL and chronic disease with environmental HRQoL were attenuated and became non-significant when the psychological variables (depression, anxiety, and stress) were entered into step 4 of the regression model, suggesting the presence of mediating effects. As shown in Table 8, the test for mediating effects found that depression was negatively associated with age (a = −0.0627, *p* < 0.05) and social relationship HRQoL (b = −1.7139, *p* < 0.01), thus synthesizing the total product of indirect relationships (a × b = 0.1075, BCa 95% CI 0.0328–0.2013, *p* < 0.01). The proportion of the mediating roles of depressed employees in the social relationship HRQoL accounted for 48.5%. However, the mediating effects for physical health and environmental HRQoL exhibited non-significance.

## 4. Discussion

### 4.1. Summary of Core Findings

The current study aimed to determine the factors associated with HRQoL scores amongst employees in a Malaysian public university. The overall HRQoL scores of the employees were high in the current study, although the subdomain HRQoL scores were much lower. The physical HRQoL scores were the highest, while environmental HRQoL had the lowest scores among the employees. The overall physical HRQoL scores were influenced by age, service duration, comorbid conditions, BMI, chronic diseases, and anxious states amongst the employees. Factors associated with psychological HRQoL were age, service duration, depression, and stress amongst the employees. Age, service duration, and chronic diseases affected employees’ social relationship HRQoL score, while the environmental HRQoL scores were associated with employees’ age, occupation type, chronic diseases, and depressive states.

### 4.2. Comparisons with the Existing Literature

#### 4.2.1. HRQoL of the Employees

The overall HRQoL reported in this study was better than the HRQoL reported among university employees in Thailand [44] and academic professors in Brazil [45]. Although the overall HRQoL scores were high in the current study, the subdomain HRQoL scores were much lower; the physical and social relationship HRQoL had higher scores—72.6 (12.9) and 72.4 (14.8), respectively—while the psychological and environmental HRQoL domains scored lower—71.5 (12.1) and 70.2 (11.9), respectively. This finding was consistent with those of a previous study from Brazil [46]. The high physical HRQoL score could have been attributed to the better health profiles among respondents in the current sample, as only a small number of employees were afflicted with comorbid conditions or chronic diseases, whereas the high social relationship HRQoL score could plausibly be attributed to the building of rapport, mentorships, service delivery, and satisfactory communications between employees and students [47].

In contrast, the relatively lower psychological and environmental HRQoL scores could be conceptualized within the framework of the organization’s structure and behavior. The revolution of tertiary education systems has put greater onus on faculties to produce globally industrialized and driven graduands within the context of neoliberal reforms. Most universities or colleges at present hold autonomy in driving their institution’s mission and in determining the faculty’s needs, a hierarchical structure, and academic leadership within departments [2]. These intensifications, together with transformations from the conventional teaching mode into digital resources, are coupled with the adoption of accelerated research activities, community outreach programs, scholarly publications, and efficient delivery systems as fundamental metrics by institutions worldwide in order to cope with national and global key performance indexes in the race against university ranking systems [2,8,11,48]. The demands of such a work cycle involving overtime have induced greater pressure on faculty members, often catalyzing adverse psychological repercussions and competitiveness in the workplace environment, yet compromising university employees’ HRQoL [11].

Comparisons between the total HRQoL domain scores in the current study with those in other literature seemed difficult. Literature that evaluated HRQoL scores amongst university employees was scarce; as such, comparisons of the overall magnitude of the current study’s HRQoL scores with those of previous works need to be interpreted with caution due to measurement variability. Studies assessing HRQoL principally use generic scales, such as the WHOQOL-BREF or SF-36, rather than condition-specific measures that are commonly adopted for patient populations with different diseases, illnesses, or disabilities [2,7,10,48]. Hence, comparisons should be made across similar populations or samples that use generic measurements for logical interpretations. However, with different studies adopting different generic scales and scoring methods across a variety of HRQoL domains, efforts to evaluate consistencies between the current study’s findings and those of the previous literature were limited.

#### 4.2.2. Covariates of Employees’ HRQoL

This study found age to be positively correlated with physical, psychological, social relationship, and environmental HRQoL. Similar findings were found in previous studies [2,44]. Such consistencies were in line with the narrative advocated by Netuveli and Blane [49], in which aging has a predominant influence on better HRQoL. As employees in the current study were mostly in the middle-aged group or beyond and within the public service of Malaysia, it could be assumed that these employees would have been accustomed to their working circumstances and advanced their career further in academia. In addition, senior employees would have self-reflected on the meaning of life due to changing roles within the working environment, prompting them to have a greater sense of belonging, happiness, and healthy behaviors, leading to better HRQoL [44]. Conversely, longer service durations significantly deteriorated employees’ physical, psychological, and social relationship HRQoL, while employees who were non-academic support staff had lower environmental HRQoL scores. These occupational attributes were corroborated well among the HRQoL domains. Although the majority of employees in the current sample were non-academic support staff who were inclined to office administration and clerical duties with long service durations, they were not necessarily guaranteed to be promoted to higher ranks within the university work environment [13]. Such systems may catalyze frustration, competitiveness, and reduced job satisfaction [13], causing decreased social relationship and environmental HRQoL. It is plausible that long durations of administrative or clerical duties are ergonomically hazardous and stressful [13,50], yet prone to deteriorating employees’ physical and psychological HRQoL.

The literature has established that overweight and obese individuals tend to impair their physical functioning and emotional wellbeing, thus compromising their HRQoL [51,52,53]. Although the current study was in line with this postulation, obese employees had significantly lower physical HRQoL scores; however, this did not impact their psychological HRQoL. Such findings were previously found in children [54] and adult populations in Australia [55]. However, Kortt and Dollery [55] proved that although BMI was not related to HRQoL, there was a significant negative relationship between BMI and psychological attributes, particularly depression and anxiety. Following this observation, the current study could somewhat advocate for a moderating effect between BMI and anxiety, as the unstandardized coefficient for BMI was largely inflated when the anxiety variable was entered in the fourth step of the regression model and became statistically significant, indicating probable correlations, yet forcing the physical HRQoL to deteriorate. Such moderation effects could also be observed for employees with multi-morbidities or who were afflicted with chronic diseases, as anxiety could be amplified due to an individual’s emotional shock from a diagnosis [56], thus indirectly deteriorating their physical HRQoL. This study found that multi-morbidities and chronic diseases negatively impacted physical HRQoL. Previous studies among Brazilian university teachers [2] and government employees in Malaysia [57] reported similar associations among co-morbid conditions, chronic diseases, and HRQoL. Diagnoses of comorbid conditions or chronic diseases have the tendency to reduce job performance, as such conditions restrict one’s physical capabilities, affect career development within an organization, and limit social interactions or inclusion [57]. These circumstances subsequently affect people’s joy and deteriorate their social relationship and environmental HRQoL, as reported in the current study. Interestingly, such situations did not deteriorate the psychological HRQoL in the current study. Two plausible explanations could be suggested here. The first is the availability of health-screening programs in Malaysian public universities for employees aged 40 years and above. Such screening programs allow routine follow-ups, early treatment, and facilitation of coping mechanisms for better emotional wellbeing. Secondly, as the majority of the employees in this sample were married, they could have obtained strong social support and a sense of concern from close confidants in order to negate negative thoughts and perceptions and catalyze positive coping mechanisms, thus reducing the deteriorating impact on psychological utility.

In line with reforms in the tertiary education sector, HRQoL has been associated with academic employability skills across faculties [58]. Academics and associate faculty members continuously struggle to equip themselves for the current transformation towards digital education and technological advancements, to acquire new skills to assist with the university’s work and administration, and to digest the continuous availability of new scientific knowledge in the scholarly literature. Such requirements have imposed greater psychological pressure on faculties worldwide, thus decreasing their HRQoL states. This study found that psychological states (depression and stress) deteriorated employees’ psychological HRQoL, while anxiety negatively impacted their physical and environmental HRQoL. Previous studies similarly concluded that psychological attributes, particularly stress, decreased the HRQoL of academics [10] and university employees [12]. The direct impacts of conflicts between colleagues, work burden, and time pressure are attributable to decreased HRQoL amongst university employees [10]. Such a work life in the environment of a university could modify relationships with family, friends, colleagues, and students or could even affect employees’ leisure time [59]. The current study found that employees’ depressive states mediated the association between age and social relationship HRQoL. Such circumstances could be explained from the theoretical perspective of cognitive appraisal, which includes primary appraisal (evaluation of a situation’s harm or benefit) and secondary appraisal (controllability and coping mechanisms) [60]. Aging could lead to negative psychological states, such as depression, in view of primary appraisal stressors, such as being ill, workplace pressures, competitiveness for promotion, or conflicts with colleagues, and in view of secondary appraisals of the ability to cope with such physical and workplace stressors. In contrast, aging could balance the harmful primary appraisal, resulting in reduced depressive states amongst employees, such as by having their vocation in their academic work life. Such psychological states may ultimately influence HRQoL through physiological or behavioral processes for the control of potential stressors.

### 4.3. Study Limitations

The limitations of this study need to be acknowledged. Firstly, the cross-sectional design of the current study was not sufficiently able to establish causal relationships between HRQoL and the tested covariates. Secondly, the employees from a single public university were not nationally representative; hence, an extrapolation of the study’s findings could not be carried out. Thirdly, the application of the common method variance—although useful for theoretical model building and identification of potential confounders, mediators, or moderators via interaction regressions—is prone to common method bias, which synthetically deflates or inflates regression weights as a consequence of variables being analyzed within the same self-reported measures [61]. Fourthly, the use of the generic HRQoL scale limited the exploration of specific occupational and health-related factors. The sample size of this study may have increased the chances of type II error in the analysis, as some variable associations reflected near statistical significance.

## 5. Conclusions

The current study found socio-demographics, risky health behaviors, health profiles, and psychological attributes to be significantly associated with employees’ HRQoL. Age was the only variable that had a statistically significant positive correlation with all four HRQoL domains. Service duration, comorbid conditions, BMI, chronic diseases, and being anxious significantly influenced physical HRQoL. The factors associated with psychological HRQoL were service duration, depression, and stress among the employees, while service duration and chronic diseases impacted employees’ social relationship HRQoL. The factors associated with environmental HRQoL were occupation type, having chronic diseases, and depressive states among the employees. Depression significantly mediated the influence of age on social relationship HRQoL. This study is the first to provide comprehensive information on factors affecting academics’ HRQoL, and it could be applied to craft appropriate interventions within the organizational structures and behaviors in order to improve the HRQoL amongst university employees.

## Figures and Tables

**Table 1 ijerph-18-10903-t001:** Sample characteristics (*n* = 397).

Characteristics	*n* (%)
** *Demographics* **	
**Gender**	
Women	250 (63.0)
Men	147 (37.0)
**Age (years), mean (SD)**	43 (7.6), min = 27, max = 60
**Marital status**	
Single	44 (11.1)
Married	353 (88.9)
**Occupation**	
Academics	95 (23.9)
Non-academics (Professionals)	75 (18.9)
Non-academics (Support staff)	227 (57.2)
**Household income level (MYR) (*n* = 290)**	
B40	78 (26.9)
M40	141 (48.6)
T20	71 (24.5)
**Service duration (years)**	
≤10	53 (13.4)
>10	344 (86.6)
** *Risky Health Behaviors* **	
**BMI (kg/m^2^)**	
Underweight (<18.5)	6 (1.5)
Normal (18.5–24.9)	129 (32.5)
Overweight (25–29.9)	159 (40.1)
Obese (≥30)	103 (25.9)
**Current smoker (*n* = 396)**	
No	366 (92.4)
Yes	30 (7.6)
**Physical activity (*n* = 391)**	
Active	162 (41.4)
Inactive	229 (58.6)
** *Health-Related Information* **	
**Comorbidities (*n* = 394)**	
≤1	364 (92.4)
>1	30 (7.6)
**Chronic diseases (*n* = 395)**	
No	383 (97.0)
Yes	12 (3.0)
** *Psychological States (DASS 21)* **	
**Depression status**	
Normal	264 (66.5)
Depressed	133 (33.5)
**Anxiety status**	
Normal	194 (48.9)
Anxious	203 (51.1)
**Stress status**	
Normal	300 (75.6)
Distressed	97 (24.4)
** *HRQoL Domain Attributes* **	
Physical HRQoL score, mean (SD)	72.6 (12.9), min = 31, max = 100
Psychological HRQoL score, mean (SD)	71.5 (12.1), min = 25, max = 100
Social relationship HRQoL score, mean (SD)	72.4 (14.8), min = 19, max = 100
Environmental HRQoL score, mean (SD)	70.2 (11.9), min = 38, max = 100
Total HRQoL score, mean (SD)	99.4 (11.1), min = 64, max = 127

**Table 2 ijerph-18-10903-t002:** Association between HRQoL and socio-demographic characteristics (*n* = 397).

Characteristics	Physical Health		Psychological		Social Relationships		Environmental	
	Mean (SD)	*p*-Value	Mean (SD)	*p*-Value	Mean (SD)	*p*-Value	Mean (SD)	*p*-Value
**Gender**								
Women	72 (12.9)		71 (12.0)		72.3 (14.9)		69.9 (11.7)	
Men	73.5 (13.2)	0.256	72.5 (12.2)	0.229	72.6 (14.6)	0.821	70.8 (12.6)	0.481
**Age (years) ^#^**	0.179	0.068	0.119	0.018	0.186	0.067	0.206	<0.001
**Marital status**								
Single	72.5 (14.9)		72.5 (14.3)		70.3 (15.5)		71.4 (12.1)	
Married	72.6 (12.7)	0.989	71.4 (11.8)	0.559	72.7 (14.7)	0.316	70.9 (11.9)	0.486
**Occupation type**								
Academics	73.1 (13.3)		72.3 (11.6)		72 (15.8)		73.4 (10.7)	
Non-academics (Professionals)	75.1 (12.2)		73.5 (11.3)		75.3 (14.2)		72.9 (10.7)	
Non-academics (Support staff)	71.5 (13.1)	0.098	70.5 (12.4)	0.144	71.6 (14.6)	0.166	68 (12.5)	<0.001
**Household income level (MYR)**								
B40	73.7 (14.0)		71.6 (11.6)		73.2 (15.3)		68.7 (12.2)	
M40	72.6 (12.4)		70.4 (12.3)		72.0 (14.1)		69.6 (11.4)	
T20	74.3 (12.8)	0.660	74.9 (10.1)	0.029	75.0 (12.2)	0.347	73.3 (10.3)	0.032
**Service duration (years)**								
≤10	75.6 (14.2)		73.8 (12.5)		77 (14.0)		70.5 (11.8)	
>10	72.1 (12.7)	0.070	71.2 (11.9)	0.141	71.7 (14.8)	0.015	70.2 (12.0)	0.852

^#^ Age was analyzed as a continuous variable—the Pearson correlation coefficient (r) is reported.

**Table 3 ijerph-18-10903-t003:** Associations between HRQoL and risky health behaviors, health-related information, and psychological states.

Characteristics	Physical Health		Psychological		Social Relationships		Environmental	
	Mean (SD)	*p*-Value	Mean (SD)	*p*-Value	Mean (SD)	*p*-Value	Mean (SD)	*p*-Value
** *Risky Health Behaviors* **								
**BMI (kg/m^2^)**								
Underweight (<18.5)	82.3 (10.1)		84.5 (6.8)		83.3 (12.9)		78.2 (10.9)	
Normal (18.5–24.9)	72.6 (13.2)		72.3 (12.2)		72.4 (14.2)		69.1 (11.4)	
Overweight (25–29.9)	74.5 (12.4)		72.8 (11.5)		73.5 (14.7)		72.3 (12.3)	
Obese (≥ 30)	69.0 (12.9)	0.002	67.8 (11.9)	<0.001	70.1 (15.5)	0.088	67.9 (11.7)	0.007
**Current smoker**								
No	75.2 (12.8)		74.0 (13.4)		75.4 (15.6)		71.7 (13.0)	
Yes	72.4 (12.8)	0.251	71.3 (11.9)	0.240	72.2 (14.8)	0.251	70.2 (11.9)	0.500
**Physical activity**								
Active	72.7 (12.8)		71.9 (11.8)		72.9 (14.1)		70.5 (12.2)	
Inactive	71.9 (13.8)	0.704	68.9 (13.2)	0.075	69.6 (18.2)	0.111	68.8 (10.6)	0.319
** *Health-Related Information* **								
**Comorbidities**								
≤1	73 (12.8)		71.7 (11.8)		72.5 (14.7)		70.3 (11.9)	
>1	66.2 (14.0)	0.006	69.6 (15.0)	0.374	69.8 (16.1)	0.339	69.7 (13.6)	0.792
**Chronic diseases**								
No	72.8 (12.9)		71.6 (12.0)		72.6 (14.6)		70.3 (11.9)	
Yes	64.3 (12.2)	0.026	66.8 (14.1)	0.175	62.5 (18.1)	0.019	66.3 (12.4)	0.249
** *Psychological States* **								
**Depression status**								
Normal	74.7 (12.1)		74 (10.1)		74.6 (13.1)		71.8 (10.8)	
Depressed	68.3 (13.7)	<0.001	66.6 (14.0)	<0.001	68 (16.9)	<0.001	67.1 (13.6)	<0.001
**Anxiety status**								
Normal	76.2 (12.5)		73.7 (11.3)		74.9 (14.0)		72.8 (10.8)	
Anxious	69.1 (12.5)	<0.001	69.4 (12.4)	<0.001	69.9 (15.2)	<0.001	67.7 (12.5)	<0.001
**Stress status**								
Normal	73.8 (12.5)		72.6 (11.2)		73.6 (13.8)		70.9 (11.3)	
Distressed	68.8 (13.7)	0.001	68.3 (13.9)	0.002	68.8 (17.1)	0.006	67.9 (13.8)	0.029

**Table 4 ijerph-18-10903-t004:** Factors associated with the physical HRQoL according to the hierarchical multiple linear regression analysis.

Characteristics	Step 1			Step 2			Step 3			Step 4		
	B	Beta	*p*-Value	B	Beta	*p*-Value	B	Beta	*p*-Value	B	Beta	*p*-Value
Age	0.186	0.109	<0.001	0.195	0.114	0.033	0.257	0.150	0.005	0.219	0.128	0.015
Non-academic professionals	2.100	0.063	0.299	1.729	0.052	0.392	1.673	0.050	0.401	0.534	0.016	0.785
Non-academic support staff	−1.522	−0.058	0.339	−1.809	−0.069	0.256	−1.872	−0.071	0.233	−2.124	−0.081	0.165
>10 years of service	−4.761	−0.124	0.020	−4.516	−0.118	0.027	−4.441	−0.116	0.027	−3.748	−0.098	0.056
Normal weight				−10.754	−0.311	0.051	−10.552	−1.947	0.052	−11.090	−0.321	0.054
Overweight				−7.198	−0.202	0.191	−7.098	−0.199	0.191	−8.262	5.226	0.115
Obese				−9.746	−0.356	0.068	−8.988	−0.328	0.088	−10.197	−0.373	0.045
>1 comorbid condition							−6.893	−0.141	0.006	−6.621	−0.135	0.006
Have chronic disease							−7.282	−0.096	0.056	−7.237	−0.096	0.049
Depressed										−2.664	−0.097	0.225
Anxious										−6.128	−0.236	<0.001
Distressed										1.294	0.043	0.574
**Model Base and Change Statistics**												
F change	3.080			1.264			6.828			10.752		
R^2^	0.031			0.037			0.070			0.142		
R^2^ change	0.031			0.006			0.033			0.072		

**Table 5 ijerph-18-10903-t005:** Factors associated with psychological HRQoL according to the hierarchical multiple linear regression analysis.

Characteristics	Step 1			Step 2			Step 3			Step 4		
	B	Beta	*p*-Value	B	Beta	*p*-Value	B	Beta	*p*-Value	B	Beta	*p*-Value
Age	0.338	0.215	0.001	0.326	0.208	0.001	0.329	0.210	0.001	0.269	0.171	0.004
Non-academic professionals	1.322	0.046	0.558	0.922	0.032	0.681	0.823	0.028	0.714	−0.809	−0.028	0.698
Non-academic support staff	−1.293	−0.054	0.556	−1.313	−0.055	0.547	−1.338	−0.056	0.540	−1.008	−0.042	0.616
B40	−0.865	−0.033	0.724	−1.633	−0.062	0.504	−1.664	−0.063	0.497	−1.077	−0.041	0.632
M40	−3.073	−0.131	0.119	−3.264	−0.139	0.096	−3.084	−0.132	0.118	−2.282	−0.098	0.208
>10 years of service	−4.972	−0.160	0.011	−4.503	−0.145	0.021	−4.386	−0.141	0.025	−4.907	−0.158	0.007
Normal weight				−10.569	−0.331	0.079	−10.644	−0.333	0.077	−8.567	−0.268	0.121
Overweight				−7.957	−0.245	0.183	−8.021	−0.247	0.180	−6.185	−0.190	0.261
Obese				−11.527	−0.462	0.048	−11.439	−0.458	0.050	−9.603	−0.385	0.073
Physically inactive				2.050	0.066	0.252	2.113	0.068	0.239	2.659	0.086	0.108
Have chronic disease							−3.901	−0.055	0.348	−2.376	−0.033	0.534
Depressed										−9.842	−0.311	<0.001
Anxious										−1.386	−0.058	0.312
Distressed										−6.600	−0.117	0.048
**Model Base and Change Statistics**												
F change	3.700			2.189			0.885			18.386		
R^2^	0.073			0.102			0.104			0.255		
R^2^ change	0.073			0.028			0.003			0.151		

**Table 6 ijerph-18-10903-t006:** Factors associated with social relationships HRQoL according to the hierarchical multiple linear regression analysis.

Characteristics	Step 1			Step 2			Step 3			Step 4		
	B	Beta	*p*-Value	B	Beta	*p*-Value	B	Beta	*p*-Value	B	Beta	*p*-Value
Age	0.231	0.119	0.025	0.232	0.120	0.025	0.255	0.132	0.013	0.201	0.104	0.052
Non-academic professionals	3.083	0.082	0.179	2.814	0.074	0.220	2.762	0.073	0.225	1.518	0.040	0.507
Non-academic support staff	−0.184	−0.006	0.919	−0.448	−0.015	0.803	−0.510	−0.017	0.776	−0.608	−0.020	0.733
>10 years of service	−6.364	−0.146	0.006	−5.953	−0.137	0.010	−5.769	−0.132	0.012	−4.911	−0.113	0.032
Normal weight				−11.645	−0.296	0.062	−11.773	−0.300	0.058	−11.700	−0.298	0.056
Overweight				−7.083	−0.175	0.257	−7.168	−0.177	0.248	−7.800	−0.192	0.203
Obese				−10.043	−0.323	0.097	−9.689	−0.312	0.107	−10.141	−0.326	0.088
Physically inactive				2.621	0.064	0.202	2.591	0.063	0.205	3.158	0.077	0.120
Have chronic disease							−10.415	−0.121	0.016	−10.265	−0.120	0.016
Depressed										−4.998	−0.160	0.052
Anxious										−2.740	−0.093	0.130
Distressed										1.770	0.052	0.511
**Model Base and Change Statistics**												
F change	3.215			1.996			5.875			4.866		
R^2^	0.032			0.052			0.066			0.100		
R^2^ change	0.032			0.020			0.014			0.034		

**Table 7 ijerph-18-10903-t007:** Factors associated with environmental HRQoL according to the hierarchical multiple linear regression analysis.

Characteristics	Step 1			Step 2			Step 3			Step 4		
	B	Beta	*p*-Value	B	Beta	*p*-Value	B	Beta	*p*-Value	B	Beta	*p*-Value
Age	0.334	0.218	<0.001	0.334	0.218	<0.001	0.345	0.225	<0.001	0.317	0.207	<0.001
Non-academic professionals	0.468	0.016	0.830	0.353	0.012	0.872	0.135	0.005	0.950	−0.338	−0.012	0.872
Non-academic support staff	−4.479	−0.192	0.036	−4.463	−0.192	0.037	−4.513	−0.194	0.034	−4.128	−0.177	0.042
B40	0.454	0.018	0.848	0.179	0.007	0.940	0.124	0.005	0.958	0.016	0.001	0.995
M40	−0.524	−0.023	0.783	−0.661	−0.029	0.729	−0.296	−0.013	0.877	−0.039	−0.002	0.983
Normal weight				−5.597	−0.179	0.337	−5.707	−0.183	0.325	−5.567	−0.178	0.315
Overweight				−3.694	−0.116	0.525	−3.790	−0.119	0.512	−3.501	−0.110	0.526
Obese				−3.878	−0.159	0.492	−3.668	−0.150	0.514	−3.716	−0.152	0.489
Have chronic disease							−8.000	−0.115	0.048	−6.778	−0.097	0.079
Depressed										−3.119	−0.101	0.119
Anxious										−4.948	−0.211	<0.001
Distressed										−4.659	−0.085	0.160
**Model Base and Change Statistics**												
F change	5.700			0.503			3.962			10.403		
R^2^	0.092			0.097			0.109			0.200		
R^2^ change	0.092			0.005			0.013			0.091		

**Table 8 ijerph-18-10903-t008:** Mediating roles in HRQoL domains.

HRQoL Domain	Mediators	a	b	a × b (BCa 95% CI)
Physical health	Depressed	−0.5671	−0.8135 *	0.4613 (−0.2698–1.7309)
	Anxious	−0.3215	−0.1165	0.0374 (−0.3986–0.6500)
	Distressed	−0.2637	−1.4121 **	0.3723 (−0.8571–1.8127)
Social relationships	Depressed	−0.0627 *	−1.7139 **	0.1075 * (0.0328–0.2013)
	Anxious	−0.0171 *	0.3444	−0.0059 (−0.0414–0.0158)
	Distressed	−0.0526 *	−0.5444	0.0287 (−0.0212–0.0894)
Environmental	Depressed	−0.9576	−1.0042 *	0.9616 (−0.9725–3.1810)
	Anxious	−0.4709	−0.4191	0.1973 (−0.7763–1.4543)
	Distressed	−0.5477	−0.6362	0.3484 (−1.0795–2.0721)

* *p* < 0.05; ** *p* < 0.01; BCa 95% CI denotes bias-corrected–accelerated 95% confidence intervals. The predictor variable for physical HRQoL was “service duration >10 years”; for social relationship HRQoL, it was “age”; for environmental HRQoL, it was “having chronic diseases.”

## Data Availability

The data presented in this study are available within the article.

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
