# Peer review of "Factors Associated with Health-Related Quality of Life amongst Employees in a Malaysian Public University"

_ijerph, 2021, doi:10.3390/ijerph182010903_

Round 1

Reviewer 1 Report

Thank you for the opportunity to review your manuscript. Below are my suggestions/questions:

1.Is the deterioration of HRQoL a conjecture or conclusion? If it is a conclusion, is there any literature or data to prove it? (row 61 to 64)

2.This paragraph clearly introduces the definition and measurement tools of HRQoL. But, it lacks a more complete explanation of why this study was conducted. (row 65 to 70)

3.Study Design lacks detailed introduction. This part simply tells the time and place. (row 92 to 95)

4.Was the questionnaire recovery 100%? Was the questionnaire screened? Were all the questionnaires collected used in the study? (row 125)

5.The manuscript did bring up an important topic. But, there is a lack of motivation for the study. After reading the introduction, I was not clear about the research purpose. And why did the author conduct the study? Please develop this part.

Author Response

Author Response to Reviewer Comments

Reviewer 1

Comment 1:

Is the deterioration of HRQoL a conjecture or conclusion? If it is a conclusion, is there any literature or data to prove it? (row 61 to 64)

Author Response:

Dear reviewer, thank you for your comment. We missed the reference. Kindly note that it has been added now.

Comment 2:

This paragraph clearly introduces the definition and measurement tools of HRQoL. But, it lacks a more complete explanation of why this study was conducted. (row 65 to 70)

Author Response:

Thank you for your comment. Since you requested to add the motivations of this study to be described, similar to comment 5 below, we have now addressed your suggestion as required, as in comment 5 below. We have added the motivation of the study under the introduction part, lines 88-100. In addition, we thought it would be useful to add another sentence on the advantages of using generic scales to be applicable for this study. This is reflected on lines 71-74. An additional reference was added.

Comment 3:

Study Design lacks detailed introduction. This part simply tells the time and place. (row 92 to 95)

Author Response:

Thank you for your comment. We have made some restructuring of the sentence to include more detailed description on the methods, study design as cross-sectional and sampling technique. The following description was added:

“An anonymous questionnaire-based survey was administered and a basic medical examination was conducted by the study team from the Universiti Kebangsaan Malaysia (UKM), Bangi, Selangor, Malaysia to address the study objectives. This was a descriptive-analytical cross-sectional single-institution study conducted between April and June 2019 via simple-random sampling method among employees in UKM.”

We hope that our editions would be satisfactory to you.

Comment 4:

Was the questionnaire recovery 100%? Was the questionnaire screened? Were all the questionnaires collected used in the study? (row 125).

Author Response:

We appreciate your queries here. The recovery (response rate) was 94.3%. This has been described under subsection 3.1 (Sample Characteristics), page 5, lines 235-236. Subsequently, we have added a new sentence on the reasons for exclusion of responses, reflected in lines 236-238.

As to the query if the questionnaire was screened or not, kindly note that we have used scales that were validated for Malaysia using the local language. Hence, we conducted face and content validity of the questionnaire based on opinions from two experts, one, a psychologist who is also an author in this study (N.B.A.K.), while another independent public health specialist who was not in the study team. The agreement of content between these two experts were concurrent. We subsequently pilot tested the questionnaire among 20 employees who were not part of the study. Additional grammatical errors or misspellings were subsequently corrected, and the final version of the questionnaire was administered to participants. 

This description is added under sub-section 2.4 (Study Instrument), last paragraph, lines 212-218. We hope that our revisions will be satisfactory to you.

Comment 5:

The manuscript did bring up an important topic. But, there is a lack of motivation for the study. After reading the introduction, I was not clear about the research purpose. And why did the author conduct the study? Please develop this part.

Author Response:

We appreciate your suggestions here. We have added the motivation of the study as follows under the introduction part, lines 88-100.

“As universities fundamentally engage to the development of the society’s socio-cultural structure, the relationship between employees HRQoL and performance are vital to the institution’s aim and success (Singh and Singh 2015). Improvement of employees HRQoL has the capability to escalate the quality of teaching practices and deliveries (Mustapha 2013), yet prompts a metric of success for an academic institution (Afsar 2015). Better HRQoL of the university employees catalyses greater productivity, commitment and performance quality of service delivery in an academic institution (Daud 2010). As employees within tertiary institutions form the foundation of the organization’s structure and function within a university, there is a need to explore the different dimensions of HRQoL and their related factors to understand how these dimensions could risk influencing their performances in universities.”

These references are cited accordingly:

Afsar, S. T. 2015. “Impact of the Quality of Work-Life on Organizational Commitment: A Comparative Study on Academicians Working for State and Foundation.” ISGUC The Journal of Industrial Relations and Human Resources 17 (2): 45–75.

Daud, N. 2010. “Quality of Work Life and Organizational Commitment Amongst Academic Staff: Empirical Evidence from Malaysia.” In International Conference on Education and Management Technology, 271–5. IEEE.

Mustapha, N. 2013. “The Influence of Financial Reward on job Satisfaction among Academic Staffs at Public Universities in Kelantan, Malaysia.” International Journal of Business and Social Science 4 (3): 244–9.

Singh, O. P., and S. K. Singh. 2015. “Quality of Work Life of Teachers Working in Higher Educational Institutions: A Strategic Approach Towards Teacher’s Excellence.” International Journal of Advance Research in Computer Science and Management Studies 3 (9): 180–6.

Reviewer 2 Report

For an international audience, please expand on grade DG, DS, 41, etc. 

Similarly, please explain what is the difference between the nonacademic professionals and staff. Some examples will help. 

Was there any analysis done to see if there was a difference in the HRQoL scores between academic, nonacademic professional, and nonacademic staff.

Author Response

Author Response to Reviewer Comments

Reviewer 2

Comment 1:

For an international audience, please expand on grade DG, DS, 41, etc. 

Author Response:

We have edited the sentence to provide more details on the grading system:

Occupation types were classified based on occupational grades as determined by the Public Education Service of Malaysia; academics (grades DS 41, 43, 44 and VK7), non-academics professionals (all categories 41 and above, except DS) and non-academics support staff (all categories below grade 41) that is determined based on service duration, degree qualifications and service performance for hierarchical promotional schemes. These editions are reflected under Section 2.4 (Study Instruments), lines 131-136.

The following additional reference has been cited too:

Public Education Service Department. EZ Skim. https://www.interactive.jpa.gov.my/ezskim/klasifikasi/klasifikasi.asp

Comment 2:

Similarly, please explain what is the difference between the nonacademic professionals and staff. Some examples will help. 

Author Response:

Regarding the difference between the non-academic professionals and staffs, the following descriptions has been added:

Non-academic professionals were those having degree level qualifications and above but are not involved in academic teaching or learning activities. This occupational group mostly serve as administrators, or employees in the engineering, information, communication and technology departments or units. In contrast, staffs with qualifications below the degree level or those involved in support services such as administration duties (such as clerks or assistant officers) are classified as non-academics support staffs. A reference has been cited as well. The changes are reflected under Section 2.4 (Study Instrument), lines 136-142.

We hope that these changes would be satisfactory to you.

Comment 3:

Was there any analysis done to see if there was a difference in the HRQoL scores between academic, nonacademic professional, and nonacademic staff.

Author Response:

Dear reviewer, yes, we did conduct an analysis using ANOVA with post-hoc Bonferroni to determine the mean difference between groups at the univariate level. This is reflected on Table 2, and the description of the results is exhibited under sub-section 3.2 (Association between HRQoL and Sample Characteristics), lines 272-276. The results showed statistically significant difference only prevailed for the environmental QOL domain at the univariate level. Hence, this variable was only selected to determine the final “predictor model” that sustained factors associated with environmental QOL in the multivariate analysis (Table 7). The results description is made available under sub-section 3.7 (Factors Associated with Environmental HRQoL by Hierarchical Multiple Linear Regression Analysis), page 15.